# Active pro-vaccine and anti-vaccine groups: Their group identities and attitudes toward science

Józef Maciuszek[1], Mateusz Polak [1]*, Katarzyna Stasiuk[1], Dariusz Doliński[2]

1 Institute of Applied Psychology, Faculty of Management and Social Communication, Jagiellonian University, Krakow, Poland, 2 Institute of Psychology, Faculty of Psychology in Wroclaw, SWPS University of Social Sciences and Humanities, Wroclaw, Poland

* mateusz.polak@uj.edu.pl

**Data Availability Statement:** All relevant data are within the manuscript and its Supporting information files.

**Funding:** The publication was funded by the Priority Research Area Society of the Future under

## Abstract

Vaccine rejection is a problem severely impacting the global society, especially considering the COVID-19 outbreak. The need to understand the psychological mechanisms underlying the active involvement of the pro-vaccine and anti-vaccine movements is therefore very important both from a theoretical and practical perspective. This paper investigates the group identities of people with positive and negative attitudes towards vaccination, and their attitudes toward general science. A targeted sample study of 192 pro-vaccine and 156 anti-vaccine group members showed that the group identity of pro-vaccine individuals is higher than of anti-vaccine individuals. and that both pro-vaccine and anti-vaccine individuals had a positive attitude toward science. Results are discussed in context of the heterogeneity of motivations causing vaccine rejection and the relation between active involvement in online discussion and group identity.

## 1. Introduction

### 1.1. Anti-science and vaccine skepticism

The achievements of modern society are built on scientific discoveries and their applications. However, in recent years, a concerning decrease in public confidence in science has been observed, and a strong anti-scientific counterculture has emerged [1]. Based on their own sources of information and selectively ignoring research that contradicts their beliefs, many laypeople believe that they have the background to challenge established scientific facts, and there is no indication that this trend will slow or reverse in the near future [2, 3]. There are several well-researched instances of anti-science. For example, although the theory of evolution is a fundamental idea of modern biology and one of the greatest achievements of Western thought, some people literally interpret the biblical Book of Genesis in asserting that God created the world and mankind within six days, and that it happened no more than 10,000 years ago [4, 5]. Moreover, despite substantial data and climatological analyses, some people deny climate change or reject the assumption that humans contribute to it [6, 7]. Similarly,

the program "Excellence Initiative – Research University" at the Jagiellonian University in Krakow. https://id.uj.edu.pl/en_GB/ The funders had no role in study design, data collection and analysis, decision to publish, or preparation of the manuscript.

**Competing interests:** The authors have declared that no competing interests exist.

although vaccination is widely considered one of the most important achievements of medicine and has saved millions of lives, vaccine skepticism is gaining increasing support. On the basis of theory of planned behavior [8–10] one can assume that vaccine skepticism strongly predicts its behavioral counterpart—vaccine rejection. People's refusal to be vaccinated or have their children vaccinated poses increasing danger not only for them but also for the population's resistance to infectious diseases. The problem of vaccine rejection is of particular importance in the current situation—the entire world is affected by the COVID-19 pandemic, and a large percentage of citizens is not willing to vaccinate against it (or hesitates to do so), putting the prospect of achieving herd immunity at risk. In Poland, where we conducted our study, only about 50% of the population was vaccinated for COVID19 as of 15% below the European average [11].

Understanding the predictors of both negative and positive attitudes towards vaccination is therefore extremely important both from a purely scientific and applied perspective. In studies conducted before the outbreak of the COVID-19 pandemic, particular attention was paid to the individual differences which might predict attitudes toward vaccines and vaccination. For example, anti-vaccine attitudes are associated with orthodox religiousness, moral purity concerns [12], conspiratorial thinking and individualistic/hierarchical worldviews [13]. We wanted to add to the existing research by taking into account the perspective of group processes. We are observing an important social phenomenon, which is of particular significance during COVID-19—the presence of highly organized anti-vaccination movements, which remain very active even during the pandemic. We assume that the functioning of vaccination opponents, as well as vaccination supporters can be viewed from the perspective of group processes. Contemporary social psychology recognizes that the key processes for group formation are identification with the group and commitment to group functioning. Group identity is seen from a cognitive-motivational perspective and the perspective of intergroup relations [14].

Based on this assumption, we decided to conduct a study of individuals who hold extremely positive or negative attitudes toward vaccination and are heavily involved in various activities (e.g., discussions, conference attendance) that align with their attitudes. The basis for referring to involvement in activities is the contemporary approach to group formation, in which involvement is treated as a fundamental process [15]. In addition to the question of defining oneself in terms of belonging to a group, contemporary analyses place a strongly emphasis on different forms of involvement in group functioning [16].

Our main goal was to investigate the group identity of active supporters and opponents of vaccination, comparing its level in both groups. We also intended to describe these oppositional groups in terms of four modes of group identity [17] (according to Roccas' theory).

The aim of our study was also to test the attitudes toward science in pro and antivaccine group. Since anti-vaccine individuals often reject the scientific evidence supporting vaccine safety and efficacy and question the voice of medical authorities (e.g., [18–20], there is a fairly widespread perception that they are anti-scientific. However, results of existing research show that that rejecting vaccinations is not necessarily the same as rejecting science in general [18–20].

We wanted to test whether attitudes toward science are significantly different in vaccination supporters and opponents, and what the relationship is between the level of group identification and attitudes toward science in the two groups. Another important object of interest for researchers of group processes is the issue of perceptions of other social groups; in our research we were interested in how these two groups perceive each other's knowledge about vaccines.

## 1.2. Group identity

The strength, certainty and clarity of attitudes is strongly influenced by perceived social support for one's attitude [21, 22]. The group to which people belong is one of the most important sources of such support. A strong sense of identity with the group is related to the intensity of contacts with its members and the exchange of information about important issues. Thus, contacts with members of one's own group become both a source of different beliefs and a factor increasing the strength and subjective rightness of one's attitudes [23, 24]. That is why we believe that the level of group identity may be related to attitudes towards vaccination in both vaccine-supporters and vaccine-rejectors. For instance, it has been demonstrated in research conducted on nurses, that their group identification (as a nurse) constitutes an independent predictor of the intention to vaccinate against influenza [25].

Roccas and her colleagues [17] emphasized that group identity is considered within different perspectives: interculturalism (individualism—collectivism), intergroup comparison, identifying with a nation (e.g. patriotism) and within professional organizations. Integrating these perspectives, they suggest that identification with groups involves four correlated but distinguishable modes: 1. Importance—concerns how much individuals see the group as part of who they themselves are. It is a cognitive type of identification; defining oneself by belonging to a group ("we" in opposition to "they"). 2. Commitment—the motivation to act for the good of the group even at the expense of one's own individual interests. Identification in this sense means a strong positive feeling for one's own group and a tendency to defend it at all costs. Commitment understood in this way is a typical component of patriotism as well as collectivism. 3. Deference—group organization based on hierarchical relations and subordination of individuals to group norms, symbols and leaders (this is characteristic of totalitarian regimes). 4. Superiority—own group is better than others (typical of nationalism and collective narcissism). Roccas et al. assumed that these four modes of identification are meaningful for large social categories, provided that they are sufficiently entitative for their members to think of them as groups entitativity is understood as "the extent to which a group is perceived as being a coherent unit in which the members of the group are bonded together in some fashion" [26 p. 131].

Roccas et al. created a questionnaire to measure these modes of identification and in their research showed that it retains a four-factor structure for both national and organizational identity. They created a questionnaire to measure these modes of identification and in their research showed that it retains a four-factor structure for both national and organizational identity.

In our project, we incorporated this model of group identity. We were interested in whether the four modes of identification are relevant to pro-vaccine and anti-vaccine groups-which are without a formal hierarchy or clear norms. Whether the identification of these groups maintains a four-factor structure and the differed modes of identification with these groups correlate positively with each other?

Group identity consists of what makes a person subjectively different as a member of his or her own group from the members of other groups (to which he/she does not belong). In principle, any domain (value, symbol, attitude) can become the basis for group identity, provided that it is accepted and considered valid by the members of the group. In accordance with Tajfel & Turner's classic view [27] we assume that group identity develops not only during face-to-face interactions but also in indirect communication (including electronic), without the necessity for direct contact with other group members in the real world. In this case, "identification is largely symbolic rather than based on interpersonal relationships" [17 p. 281].

Opinion-based groups often form around controversial issues, while oppositely defined opinion groups tend to advocate contrary viewpoints on these issues. The social identity of both opposite groups is then defined by the shared in-group beliefs [28]. From the social-psychological point of view, vaccine rejecters and vaccine supporters can be described as social groups which center around a shared opinion, rather than more tangible characteristics such as gender, nationality or affiliations [29]. We assume that vaccine supporters base their identity on majoritarian common sense, concern for the common good (i.e. health) and respect for scientific authorities. It is difficult to assume a single leading factor in the group identity of vaccine opponents. A variety of factors may influence their involvement: business and political goals, individuals' characteristics and dispositions (e.g., heightened conspiratorial thinking), parents' fears for their children, etc. Additionally, anti-vaccinationists are probably aware of their group's heterogeneity in terms of beliefs; for they differ in their attitudes toward science, their assessment of the existence of coronavirus and the real threat of a pandemic, the content and intensity of conspiracy thinking These factors might lead to a higher level of social identity in the pro-vaccine group than in the anti-vaccine group, even though the minority usually has a stronger group identity than the majority.

## 1.3. Are vaccine rejecters generally anti-scientific?

One may assume that one of the factors that differentiate vaccine supporters and vaccine opponents is the attitude towards science and scientific knowledge. Scientific knowledge is based on causal relationships between variables and on estimation of the probabilities of the occurrence of particular events. The belief in the effectiveness of vaccines is evidently based on such evidence-based data. Therefore particular, rare cases of post-vaccination negative reactions (like fever or chills) or correlational (not causal!) relationships between vaccination and autism are not enough for those who believe in science to consider vaccination harmful or not necessary. In contrast, common-sense knowledge is based on the observation of single events and on assigning them greater importance than evidence-based data. Therefore, people who do not believe in science may attach greater importance to single cases, correlation relationships, as well as conspiracy theories. Our aim was to examine the role of attitudes toward science in the construction of group identity of vaccination supporters and opponents.

It is often mentioned in the literature that anti-vaccine individuals tend to reject scientific evidence supporting the safety and efficacy of vaccines, and question the legitimacy of relevant research, expertise and medical authority [18–20]. One may think, therefore, that vaccine rejection is strongly associated with general anti-scientific attitudes. The true picture is much more complicated, however. As Rutjens and his colleagues demonstrated [12, 30], science skepticism is not a homogenous phenomenon. They found that vaccine rejection correlates only moderately with other domains of science skepticism (such as climate change denial or GMO rejection), and there is much more to vaccine hesitancy than a general rejection of science.

In addition, leaders of anti-vaccine movements often bolster their authority with scientific titles or medical credentials. One should also take into account that language analysis of anti-vaccination comments published online showed that they contain linguistic markers of analytical thinking, with logically structured statements that mimic valid scientific information [31], which further indicates that vaccine rejection is not associated with general anti-scientific attitudes.

Anti-science, as well as vaccine hesitancy, have long been the subject of substantial research. Particular attention was paid to the individual differences which might predict attitudes

toward science. While appreciating the value of such approach, we propose to supplement it with the perspective of social psychology—taking into account group processes, and particularly group identity.

## 1.4. Current research

The research presented in this paper was aimed at investigating the associations between group identity and attitudes toward science. We wanted to investigate not just pro-vaccine and anti-vaccine individuals, but specifically only those who are involved in discussion about whether vaccines are good or bad—these individuals are the most likely to have a group identity associated with their attitudes toward vaccination, and most likely to have 'true' strong attitudes, rather than just declarations. Primarily, we wanted to see whether the pro-vaccine and anti-vaccine groups have similar identity profiles. We also wanted to see whether active anti-vaccine group members actually are anti-scientific, and whether pro-vaccine group members support science. Finally, we wanted to investigate whether attitudes toward science are related with group identity and the level of involvement, which may indicate that attitudes toward science are an important motive for group activity in one or both of these groups. Additionally, it seems interesting to investigate how these two groups perceive each other's knowledge about vaccines.

As indicated in the previous section, many opponents of vaccination do not necessarily reject science, though others may exhibit science skepticism. Vaccine supporters, however, seem much more homogeneous in their acceptance of science. It can therefore be assumed that a common attitude toward science may, among other variables, serve as a solid basis of group identity for vaccination supporters, but it is less likely for vaccination rejecters. As anti-vaccine attitudes arise from various different motivations and beliefs [12, 13, 30, 32], it can be hypothesized that the social identity of vaccination rejecters will be weaker than that of vaccination supporters.

An additional question worth investigating from the perspective of group processes is how both groups evaluate each other's knowledge about vaccines and vaccination. Basic rules of social cognition allow assuming that both anti-vaccine and pro-vaccine groups would have a low opinion of the knowledge of "the other" (since these groups have antagonistic beliefs), but the underlying reasoning may differ. Pro-vaccine individuals are expected to believe that anti-vaccine individuals derive information (or rather misinformation) from unscientific and unreliable sources and that they are subject to cognitive error [33, 34]. In turn, vaccine rejecters are expected to be convinced that pro-vaccine individuals base their knowledge on manipulated and biased data, also making it unreliable [35].

In summary, the above considerations allow us to state four main hypotheses. We predicted that supporters of vaccination would have a significantly more positive attitude toward science than members of the anti-vaccine group. The assumption that trust in science is in line with supporting vaccination (a scientifically proven medical preventative measure)—and therefore being a member of the pro-vaccine group, is the basis for the second hypothesis, that in the pro-vaccine group there would be a stronger positive correlation between the levels of group identity and acceptance of science than in the anti-vaccine group. On the other hand, research indicates the motives for belonging to the anti-vaccine group may be more diverse. Thus, we expected that in the pro-vaccine group, the level of social identity would be higher than in the anti-vaccine group. Finally, in-group favoritism and outgroup negativity [36] may serve as support for the assumption that both the pro-vaccine and anti-vaccine groups would provide low evaluations of each other's knowledge about vaccines and vaccination.

## 2. Method

The study was conducted using the Ariadna Nationwide Research Panel, a Polish counterpart of mTurk—a company specialized in polling of large samples for the purpose of research. The panel enables random selection of the sample from among 100,000 registered and verified users. Additionally, it has been awarded certificates issued by recognized organizations associated with social research companies (including ESOMAR). For participation in the survey, respondents received credit points that they could exchange for gifts. Study data has not been made publicly available. We sought to survey individuals with strong pro-vaccine or anti-vaccine beliefs who also actively participate in various forms of discussion about vaccines, meaning that they have contact with other similar people and are therefore members of a group rather than lone individuals with particular beliefs. Research was approved by the Ethics Committee of the Institute of Applied Psychology at the Jagiellonian University. Electronic consent was obtained from all participants.

### 2.1. Participants

Three hundred and fifty people (203 women and 147 men), aged 18–76 years (M = 41.63, SD = 14.56), were recruited from the general population. This targeted sample was preselected from a general, representative population sample of N = 11579 based on two selection criteria: (1) are pro-vaccine or anti-vaccine and (2) are actively involved in the discussion about vaccination.

The first criterion was based on two questions: (a) *What is your opinion about vaccination* (possible answers: *You should vaccinate*; *You should not vaccinate*; and *I am not sure/I don't care*);(b)*Would you get vaccinated if there were a vaccine available for a new dangerous disease*? (answers ranging from 0 –Definitely not to 10 –Definitely yes). Participants who answered (a) *You should vaccinate* and (b) at least 7 = yes were considered pro-vaccine. Participants who answered (a) *You should not vaccinate* and (b) no more than 3 = no were considered anti-vaccine. All others were dropped from the study.

The second criterion was based on the question (c) *Do you take active part in the discussion about the need to vaccinate or the consequences of vaccination*? (yes or no), which was followed by (d) a list of various types of active discussion contexts, from which the participants could select multiple responses: Internet forums, social media, conferences, pro-vaccine/anti-vaccine societies, discussion with acquaintances and other. Participants who indicated that they took part in discussions in any of these ways were included in the sample, and all other participants were omitted.

In the final sample (N = 350), one hundred ninety-three of the participants indicated that they were pro-vaccine (i.e., they answered *You should vaccinate* to the question *What is your opinion on vaccination*), and the remaining 157 were anti-vaccine (i.e., they answered *You should not vaccinate*; please note that these participants were selected from a larger representative sample to directly investigate pro-vaccine and anti-vaccine groups). When asked an additional question, '*Would you get vaccinated against a new dangerous disease*?'(rated on a scale from 0 to 10, with 0 = *strongly disagree* and 10 = *strongly agree*), pro-vaccine individuals strongly agreed (M = 9.26, SD = 1.32) and anti-vaccine individuals disagreed (M = 2.92, SD = 3.08). These groups did not differ in terms of education level ($\chi^2$(5, N = 350) = 3.204, p = .669, Cramer's V = .096), but the pro-vaccine group was more gender-equal (97 women and 96 men vs. 106 women and 51 men in the anti-vaccine group, $\chi^2$(1, N = 350) = 10.584, p < .001, Cramer's V = .174) and had more participants residing in rural areas (pro-vaccine: 69 rural, 65 town, 59 city vs. anti-vaccine: 27 rural, 60 town, 70 city, $\chi^2$(2, N = 350) = 15.979, p < .001, Cramer's V = .226).

## 2.2. Materials and procedure

The survey was conducted online and consisted of three main parts. Part one collected demographic data (age, sex, education, residence) and responses to the above mentioned selection questions. Part two used Roccas' group identity questionnaire [17], which consists of 16 questions measuring Importance, Commitment, Superiority and Deference with respect to the participants' own group (pro-vaccine or anti-vaccine, respectively). Sample items include *Belonging to this group is an important part of my identity* (Importance); *I like to help this group* (Commitment); *Compared to other groups of this kind, this group is particularly good* (Superiority) and *It is disloyal to criticize this group* (Deference). Please see Roccas' paper for full details on the questionnaire items. The total Cronbach's alpha reliability score was α = .92. Cronbach's alphas for the four subscales were α = .91 for Importance, α = .92 for Commitment, α = .89 for Superiority and α = .86 for Deference. Similarly to the original research by Roccas [17], the modes of identification with groups strongly correlated with each other, ranging from r = .79 for Importance x Superiority to r = .89 for Importance x Commitment. Correlations of the four modes with the total identity measure ranged from r = .92 to r = .94 (all at p < .001). In fact, the observed correlations were higher than in Roccas' research, indicating that the group identities of active pro-vaccine and anti-vaccine individuals may be equally loaded by all modes.

We then ran a confirmatory factor analysis (AMOS, Maximum Likelihood estimation) to test whether the factorial structure of Roccas' group identity scale holds in our sample. The model fit was acceptable with GFI = .902, CFI = .966, RMSEA = .07 (with p < .001), and $\chi2/df$ = 2.78, therefore corroborating the original factorial structure. We also ran a CFA using a two-factor model proposed by Roccas, wherein Importance and Commitment constituted one factor, and Superiority and Deference were the second factor. The model fit was worse with GFI = .887, CFI = .957, RMSEA = .08 (p < .001) and $\chi2/df$ = 3.16. A single-factor model was still worse with GFI = .837, CFI = .932, RMSEA = .10 (p = .001) and $\chi2/df$ = 4.35. Comparing the three models using the Bayesian Information Criterion yielded BIC = 639.89 for the 1-factor model, BIC = 518.77 for the two-factor model and BIC = 495.26 for the four-factor model, indicating that the four-factor model proposed by Roccas had the best fit in our sample.

Finally, we asked participants to evaluate two statements (on a scale of 0–10 from Strongly disagree to Strongly agree) measuring their attitude toward science: *We can only rationally believe in what is scientifically provable* and *Science is the most efficient means of attaining the truth* [32]. The Cronbach's alpha reliability score was .82.

The perceived scientific basis of the outgroup's knowledge of vaccines was measured with two statements rated on a 0–10 scale from Strongly disagree to Strongly agree: *[they] base their knowledge about vaccines on scientifically confirmed information* and *[their] knowledge about vaccines, their content, usage and side effects, is significant*. The Cronbach's alpha reliability score was .88.

## 3. Results

### 3.1. Pro-vaccine and anti-vaccine group identities

We compared the group identities of the pro-vaccine and anti-vaccine groups using MANOVA. The pro-vaccine group had a stronger identity than the anti-vaccine group in all respects. All analyses were conducted in SPSS. The results are presented in Table 1 and Fig 1.

We also found that group identity positively correlated with attitudes toward science in both the pro-vaccine and anti-vaccine groups (!). In the pro-vaccine group, Superiority, Deference and the Total score also correlated with the perceived basis of outgroup knowledge about

**Table 1. Group identity levels of the pro-vaccine and anti-vaccine groups.**

| Identity | Pro-vaccine | Anti-vaccine | $F(1,348)$ | $P$ |
|---|---|---|---|---|
| Importance | M = 5.40, SD = 1.11 | M = 4.64, SD = 1.35 | 32.94 | < .001 |
| Commitment | M = 5.51, SD = 1.09 | M = 4.74, SD = 1.27 | 51.44 | < .001 |
| Superiority | M = 5.23, SD = 1.19 | M = 4.70, SD = 1.29 | 25.11 | < .001 |
| Deference | M = 5.19, SD = 1.16 | M = 4.61, SD = 1.34 | 29.28 | < .001 |
| Total | M = 5.33, SD = 1.06 | M = 4.67, SD = 1.22 | 29.45 | < .001 |

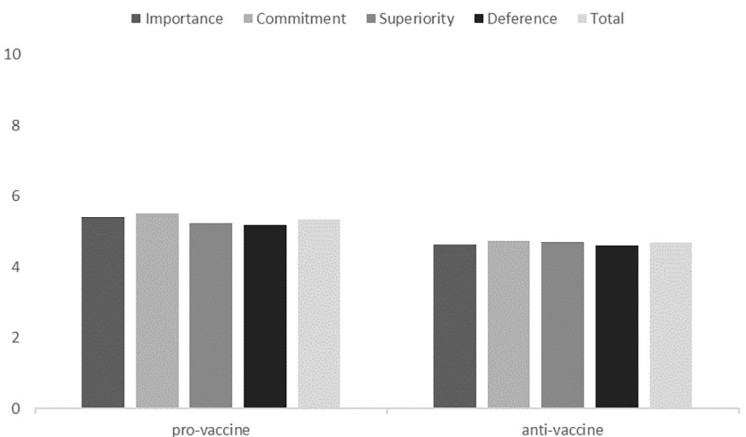

**Fig 1. Group identity levels of the pro-vaccine and anti-vaccine groups.**

vaccines, while in the anti-vaccine group, there were no significant correlations in this regard. The results are presented in Table 2. Additionally, attitude toward science did not significantly correlate with perceived basis of outgroup knowledge about vaccines ($\tau$ = -.06, p = .246).

We then ran a multiple regression analysis to investigate which of these four identity factors (Importance, Commitment, Superiority and Deference) are most strongly associated with attitudes toward science in the pro-vaccine and anti-vaccine groups. For the pro-vaccine group, the four factors explained 9% of the variance ($R^2$ = .09, F(4,188) = 4.662, p = .001) and the only significant factor was Importance ($\beta$ = .418, p = .032). For the anti-vaccine group, the four factors explained 8.6% of the variance ($R^2$ = .086, F(4,152) = 3.575, p = .008) and the only significant factor was Superiority ($\beta$ = .510, p = .001). When analyzing both groups within a single

**Table 2. Group identity correlations with attitudes toward science and perceived outgroup knowledge.**

| Identity | Attitude toward science | | Perception of outgroup knowledge | |
|---|---|---|---|---|
| | Pro-vaccine | Anti-vaccine | Pro-vaccine | Anti-vaccine |
| Importance | .289** | .160** | .055 | -.022 |
| Commitment | .270** | .179** | .048 | -.080 |
| Superiority | .191** | .247** | .152** | -.070 |
| Deference | .173** | .145* | .148** | -.063 |
| Total | .240** | .214** | .104* | -.059 |

Note:

** indicates correlations significant at p < .005,

* indicates correlations significant at p < .05

**Table 3. Types of active involvement of pro-vaccine and anti-vaccine group members.**

| Type of involvement | Pro-vaccine N | Anti-vaccine N | $X^2$ | P |
|---|---|---|---|---|
| Internet forum | 84 | 58 | 1.56 | .21 |
| Social media | 58 | 37 | 1.84 | .18 |
| Societies | 15 | 36 | 15.98 | < .001 |
| Conferences | 15 | 18 | 1.38 | .24 |
| Discussing with acquaintances | 146 | 101 | 5.34 | .021 |
| Other | 2 | 4 | n/a | n/a |

multiple regression, it turned out that the four factors explained 9% of the variance ($R2 = .090$, $F(4,345) = 8.480$, $p < .001$), and the only significant factor was still Superiority ($\beta = .216$, $p = .036$).

## 3.2. Involvement in discussion and group identity

The types of involvement in discussion did not significantly differ between the pro-vaccine and anti-vaccine group members; the pro-vaccine group members participated, on average, in $M = 1.65$, $SD = .87$ active forms of discussion, while the anti-vaccine members participated in $M = 1.59$, $SD = .99$; $t(348) = .578$. The analytical results for the individual types of active involvement are presented in Table 3; the only significant differences concerned pro-/anti-vaccine societies and discussions with acquaintances. The two groups did, however, differ in their declared level of involvement in the discussion—the pro-vaccine group reported higher involvement ($M = 8.17$, $SD = 1.36$ vs. $M = 6.91$, $SD = 1.91$; $t(273,741) = 6.94$, $p < .001$).

The declared level of involvement significantly positively correlated with all modes of group identity. We used Kendall's tau-b correlation coefficients due to violations of normality in the data; however, Pearson correlations indicated the same effects. The correlations are presented in Table 4.

## 3.3. Attitudes toward science in the pro-vaccine and anti-vaccine groups

A comparison of attitudes toward science between the groups indicated that members of the pro-vaccine group were more pro-scientific than members of the anti-vaccine group ($M = 8.02$, $SD = 1.85$ vs. $M = 6.21$, $SD = 2.80$, $t(259,721) = 6.964$, $p < .001$). Both groups presented a pro-scientific attitude (i.e., above the 5.5 neutral value), with $t(192) = 18.92$, $p < .001$,

**Table 4. Correlations between declared level of involvement and group identity.**

| Involvement correlation with | Group identity $\tau$ | | |
|---|---|---|---|
| | Total sample | Pro-vaccine | Anti-vaccine |
| Importance | .396** | .440** | .280** |
| Commitment | .466** | .452** | .399** |
| Superiority | .398** | .446** | .307** |
| Deference | .396** | .415** | .328** |
| Total | .433** | .454** | .350** |

Note:

** indicates correlations significant at p < .005,

* indicates correlations significant at p < .02.

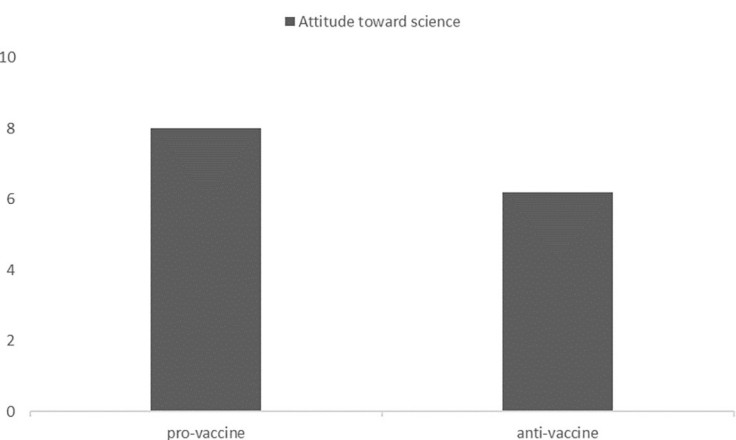

**Fig 2. Attitudes toward science in the pro-vaccine and anti-vaccine groups.**

in the pro-vaccine group and $t(156) = 3.16$, $p = .002$, in the anti-vaccine group. Results are presented in Fig 2.

We also investigated how declared active involvement correlated with attitudes toward science. In the pro-vaccine group, involvement positively correlated with pro-scientific beliefs ($\tau = .20$, p = .001), while in the anti-vaccine group, there was no significant correlation ($\tau = .10$, p = .091).

### 3.4. Perceived vaccine-related knowledge basis of outgroups

We also wanted to investigate how the pro0vaccine and anti-vaccine groups perceive each other's knowledge about vaccines, i.e. whether it is perceived as based on science or not. We found no significant differences in how these two groups perceived each other's knowledge ($t(348) = .399$, $p = .69$). These perceptions of the basis of outgroup knowledge about vaccines were below the neutral value of 5.5 (on a scale from 1 –nonscientific to 10 –scientific): $t(192) = -5.93$, $p < .001$, for the pro-vaccine group's responses and $t(156) = -4.79$, $p < .001$, for the anti-vaccine group's responses. Results are presented in Fig 3.

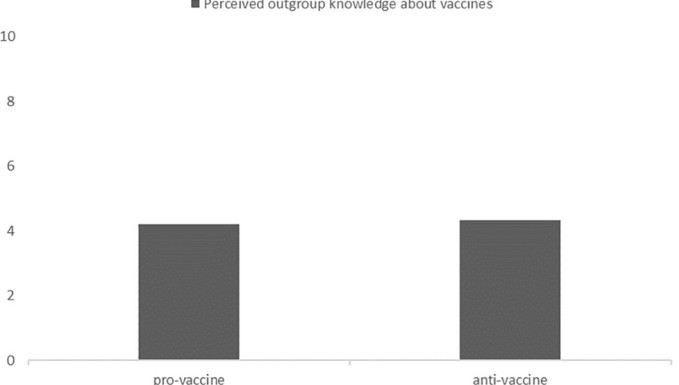

**Fig 3. Perceived outgroup vaccine-related knowledge bases (higher = more scientific).**

## 4. Discussion

Studies on the attitudes of opponents of vaccination and the motives driving them to not vaccinate their own children have so far been analyzed in a way detached from the perspective of group processes [19, 37, 38]. Our attempt sought to account for this context. Vaccination opponents and supporters interact with each other both within and between groups (usually online) and influence each other. Our understanding of the attitudes and motives of opponents of vaccination can therefore become more complete when both groups (i.e., opponents and supporters of vaccination) are analyzed simultaneously.

Let us remember that being actively involved in the discussion about vaccination was the criterion for selecting people into the study groups (in addition to being pro-vaccine or anti-vaccine). Involvement in discussions on vaccination positively correlated with group identity both in the pro-vaccine group and the anti-vaccine group. This means that active participation in such discussions was a good criterion for selecting people who can be considered members of the group.

In line with our assumptions, the pro-vaccine group had a stronger overall identity than the anti-vaccine group. It should be stressed that all modes of identification (Superiority, Commitment, Importance and Deference) were significantly stronger in the pro-vaccine group. It turned out that in the group of vaccination supporters, group identity quite strongly positively correlated with attitudes toward science. Interestingly, a positive relationship between level of identification and acceptance of science also occurred in the anti-vaccine group. It should be noted that in the pro-vaccine and anti-vaccine groups, the correlation between positive attitude toward science and the overall level of social identity was similar ($\tau$ = .240 vs. $\tau$ = .214, respectively).

The results of our research show that pro-vaccine individuals hold a strongly positive attitude toward science and are significantly more pro-scientific than anti-vaccine individuals. The result for vaccination supporters is consistent with our assumption that their activity is at least partially based on belief in science. However, the fact that anti-vaccination group members have less positive attitudes toward science than pro-vaccination group members does not mean that these attitudes are clearly negative. On the contrary, most participants who declared that they are against vaccination also agreed with the statements that it is rational to believe only what is scientifically proven and that science is the most effective means of finding truth. This result supports the assumption that vaccine rejection does not necessarily mean rejection of science, and there are different reasons for being skeptical about vaccines [12, 30]. This pattern of results challenges the previous literature in which vaccine rejection has often been treated as a manifestation of anti-science [3, 20] because although rejecting vaccines is a form of rejecting something scientifically proven, it is not equivalent to rejecting science in general. A very important result is that while positive attitudes toward science were most strongly associated with their group's Importance, the anti-vaccine group exhibited the strongest association between pro-science and Superiority. This may indicate that pro-vaccine individuals are pro-scientific because they feel that their pro-scientific group is important to them, or they feel that being pro-vaccine is important because it fits their pro-scientific attitudes. In case of the anti-vaccine group, it seems that they exhibit pro-scientific attitudes because they believe that being anti-vaccine is actually scientifically correct—the anti-vaccine group is, in their eyes, superior in understanding science. This is in line with existing research indicating that anti-vaccine groups often use excerpts from scientific papers as support for their ideas. Anti-vaccinationists can build their social identity on the belief that they possess unique, hard-earned knowledge about the true nature of vaccines—knowledge that is unavailable to the majority of the public, who are 'misled by pharmaceutical companies and their bribed physicians'. As

Motta, Callaghan and Sylvester noted [39], this overconfidence is linked to opposition to the policy of mandatory vaccination.

We also compared attitudes toward science based on the extent to which individuals in the two groups considered themselves active in discussions about vaccination. The results indicated that in the pro-vaccine group, involvement positively correlated with pro-scientific beliefs, while in the anti-vaccine group, there was no significant correlation. This result supports our assumptions that in the case of vaccination supporters, a positive attitude toward science is one of the motivations for activity, and this activity (involvement in discussions) goes hand in hand with a sense of identification with the group. In the case of opponents of vaccination, the attitude toward science, as we assumed, is not the main motive for engaging in group activity; hence, there are no significant correlations between attitude toward science and group identity.

It seems that the motive driving supporters of vaccination to actively engage in the group is consistent and is connected with, among other things, support for science and its discoveries; with trust in modern, conventional medicine; and with the perception of threats to the population due to infections carried by non-vaccinated people. In contrast, the factors motivating group membership among those who oppose vaccination are more diverse, therefore they have a weaker group identity. The minority group usually has a stronger group identity than a majority group [27]; especially with regard to uncontrollable characteristics (e.g. skin color, nationality). It turned out that this effect did not apply to the anti-vaccine group. This may be because people rarely think of themselves in terms of being pro-vaccine vs. anti-vaccine. If it happens to them (e.g. in the course of internet discussions), pro-vaccinationists see the homogeneity of their group (everyone believes in science) which strengthens the sense of group identity, while anti-vaccinationists see the heterogeneity of their group (some of them believe in science, others do not; some are afraid of epidemics, others are not; some believe in conspiracies, others do not).

We assumed (and it is consistent with the presented results) that a clearly and unequivocally positive attitude toward science is a common and important factor of support for vaccines and that the group of vaccine rejecters is more heterogeneous in this respect. Some vaccine rejecters support science, others are skeptical of it, and still others reject it [31]. However, in our research we did not find a factor which would clearly unite this group. These results are in line with existing research on vaccine hesitancy. For instance Kahan [40] showed that anti-vaccine attitudes have a global nature, without a single reason behind them, nor are they associated with particular political views, religiosity, science comprehension and their anti-scientific beliefs (p. 47). Atwell et al. [41] and Reich [42] indicate that vaccine rejecters simply believe there to be a 'healthier alternative' in form of natural medicine or similar practices, not necessarily linked to anti-scientific beliefs per se.

The current study also assessed how both groups evaluate outgroup knowledge about vaccination. The results revealed that anti-vaccine individuals evaluated the knowledge of pro-vaccine individuals negatively, and vice versa. Apart from ingroup favoritism and outgroup negativity, these results can be explained by a kind of lay epistemology that Ross and Ward [43] call naïve realism. According to them, naïve realism is based on an individual's belief that his or her opinions and preferences (here, about vaccination) stem from an objective, impartial review of facts and evidence. If someone disagrees with these (here, pro-or anti-vaccine) opinions, it means that he or she has no access to the true facts, is unable to evaluate them objectively, or is biased and blinded by his or her own interests or ideology.

Based on our research and referring the paper by Rutjens et al. [30, p. 23], we believe that the statement "science does not speak with a single voice" can be recast as "vaccine skepticism does not speak with a single voice" to reflect attitudes toward science. In other words, there

may be various types of anti-vaccine individuals—those who reject science in general and those who reject vaccines for other reasons while accepting the scientific method and published results [12, 30]. These 'other' motivations may include an egocentric strategy of relying on collective immunity without vaccinating, to protect one's family from even the slightest side effects (or even pecuniary costs) of vaccines.

The presented research has some limitations, stemming both from theory (or lack thereof) and difficulties reaching active anti-vaccine individuals. Primarily, the sample could be larger, however the 350 people in our study were the most we could find in a representative group of 11 000 participants. Particularly hard to find were active anti-vaccine individuals, which stems from the rather low percentage of anti-vaccine individuals in the population. Another serious limitation is that the associations between group identity and vaccine rejection are strongly based on attitudes toward science, which is only one of many possible reasons to become a pro-vaccine or anti-vaccine activist. This, however, is representative of the fact that we know relatively little about the reasons behind vaccine rejection, far from any unified theory, and increasingly more evidence shows that vaccine rejection may have different causes than other anti-scientific beliefs.

Summing up the results of our research, we would like to state that an approach based on the analysis of group processes allowed us to identify several important things. First, as we have mentioned above, vaccination supporters believe in science more than opponents of vaccination, but the latter group is not as homogeneous as is usually perceived. Second, contrary to widespread opinion, the group identity of vaccination opponents turned out to be weaker than that of vaccination supporters. These results show the prospective empirical value of adding an approach based on group processes to the analysis of anti-vaccine attitudes. It is, for example, extremely interesting what role online discussion with people who have different opinions about vaccination plays in shaping group identity. Similarly, it seems important to investigate whether such discussion strengthens pre-existing attitudes on both sides of the dispute or causes a more careful analysis of the respective arguments. These are only examples of questions worth answering by conducting research on anti-vaccine attitudes and behavior based on the group processes perspective.

## Supporting information

**S1 Database.**
(SAV)

## Author Contributions

**Conceptualization:** Józef Maciuszek, Katarzyna Stasiuk, Dariusz Doliński.

**Data curation:** Mateusz Polak.

**Formal analysis:** Mateusz Polak.

**Funding acquisition:** Józef Maciuszek.

**Investigation:** Dariusz Doliński.

**Methodology:** Józef Maciuszek, Mateusz Polak, Katarzyna Stasiuk.

**Supervision:** Józef Maciuszek, Dariusz Doliński.

**Visualization:** Mateusz Polak.

**Writing – original draft:** Józef Maciuszek, Mateusz Polak, Katarzyna Stasiuk, Dariusz Doliński.

**Writing – review & editing:** Mateusz Polak.

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
