## [Decision Letter · Decision Letter 0]

19 Apr 2021

PONE-D-21-02562

Active pro-vaccine and anti-vaccine groups: their group identities and attitudes toward science.

PLOS ONE

Dear Dr. Polak

Thank you for submitting your manuscript to PLOS ONE. After careful consideration, we feel that it has merit but does not fully meet PLOS ONE’s publication criteria as it currently stands. Therefore, we invite you to submit a revised version of the manuscript that addresses the points raised during the review process.

I have read the ms and the feedback provided by three expert reviewers. All Rs found merit in this research but they also raise important issues that prevent me from accepting the ms in its present state. A particular opportunity here is that R2 is a qualitative researcher and this is an opportunity for you to make sure that the concepts and language is accessible even to those who are not well versed in some aspects of your study. R1 and R2 are concerned that the rationale is under-developed and does nor motivate the hypotheses. R2 also points out that the abstract needs to better represent the study. R3 on the other hand invites you to consider the role of group membership and the role it plays more prominently. 

In addition to these I would like you to consider:

1. Language and style needs some work

2. Please elaborate on Ariadna Nationwide Research Panel, e.g. where, which country, public daya or not?

3. Provide full list of items as supplement if you have not done so already.

4. Detail the software you used

5. Figures appeared to be directly copied and pasted from SPSS output? While this is not a major issue it gives the manuscript a rahter naive (for lack of a better description). Consider editing the format. You may find other prreviosuly published papers useful for ideas.

6. More importantly, consider running additional analyses to determine which aspect of group membership is more influential on attittudues toward science and vaccination. I would even consider a more complex model where dimensions of identity predict attitudues toward vaccination via attitudues toward vaccination. 

7. Please elaborate on the limitations of your research. This is important.

I would like to reiterate that my decision on the revised version of the ms will depend on how well you would be able to adress issues raised by the Reviewers and my own observations I detail above.

We look forward to receiving your revised manuscript.

Kind regards,

Huseyin Cakal

Academic Editor

PLOS ONE

Journal Requirements:

2. Please amend your manuscript to include your abstract after the title page.

Reviewers' comments:

Reviewer's Responses to Questions

**Comments to the Author**

1. Is the manuscript technically sound, and do the data support the conclusions?

Reviewer #1: Partly

Reviewer #2: Yes

Reviewer #3: Yes

2. Has the statistical analysis been performed appropriately and rigorously? 

Reviewer #1: I Don't Know

Reviewer #2: I Don't Know

Reviewer #3: Yes

3. Have the authors made all data underlying the findings in their manuscript fully available?

Reviewer #1: Yes

Reviewer #2: Yes

Reviewer #3: Yes

4. Is the manuscript presented in an intelligible fashion and written in standard English?

Reviewer #1: Yes

Reviewer #2: Yes

Reviewer #3: Yes

5. Review Comments to the Author

Reviewer #1: In the manuscript entitled “Active pro-vaccine and anti-vaccine groups: their group identities and attitudes toward science”, the authors make use of a sub-sample of 350 participants to examine attitudes towards science, social identification, and outgroup attitudes among people active in discussions pro- or against vaccines. It provides several interesting correlations between these variables, shedding light on the similarities and differences between vocal supporters and opponents of vaccines.

I found the overall argumentation and structure of the article rather difficult to follow. Although several aims and hypotheses were described, I did not find a clear research question in the manuscript. Additionally, there were some inconsistencies between hypotheses, measures, and the reported findings; most notably around the concept of social identity. I will elaborate on these concerns below.

Regarding the use of social identity in the current manuscript, some things did not become clear to me. My main concern is that the manuscript does not provide a clear reason why and how group identity/social identity should play a role among vaccine supporters and opponents. I am sure there are plenty of good reasons to study group processes here, but without a clear reason how it needs to be studied, I found it difficult to assess whether the methods were appropriate to do so. Could, for example, being personally pro- or anti vaccines not explain the same patterns in support for science?

The methods to assess social identification match the useful distinction between the four aspects of group identity (p.4). However, two things are unclear or inconsistent here. First, the hypotheses mention “level of group identity” as a singular measure, whereas the correlations are presented for each of the four aspects of group identity. Moreover, the single reliability coefficient in the measures section suggests that indeed all 16 items are considered as a single group identity scale. When considered separately, I think a factor analysis would add to the understanding of these four sub-scales. Second, the methods section lacks sample items for the identity scale. Based on the data set, I could not fully assess what the group identity exactly pertains clearly to in the items. I think this is especially important given my earlier point that it is unclear what the group identity adds in explanatory power over and above personally being pro- or anti vaccines. In the theory, I also would have liked to a see a specific discussion on this: what does a pro- or anti vaccine identity look like exactly?

I think the sub-sample drawn from the larger representative sample is fitting to test the hypotheses. The selection criteria are well-argued in the methods section. Perhaps the introduction could have been clearer that the current study concerns active/involved opponents and supporters of vaccines. Based on vaccination rates, vaccine supporters can either be an overwhelming majority group or form a minority and this can influence identification processes. Some more background information on this would help argue for the current hypotheses: how do the opponents and supporters compare to the general representative sample that they were drawn from? And what can be said beforehand about pro- and anti-vaccine groups in the current context, both practically and theoretically?

The methods and results sections contained some other puzzling information. I found the types of involvement very useful in order to get a better understanding of who the participants in the study were. However, the type of involvement was not mentioned in the introduction, hypotheses, nor methods and therefore came as a surprise to me. In the results and discussion, involvement seems to be discussed as a predictor. Additionally, the “outgroup identity” in Table 2 was not clear to me. I did not find this measure in the methods section.

For the general methodological approach of the manuscript, I think the language in the abstract, introduction, and hypotheses is not always consistent. For example, the abstract mentions “mechanisms”, suggesting group identity to potentially play a mediating role that is not test. On page 2, “understanding the predictors of […] attitudes towards vaccination” is mentioned, suggesting at least some kind of inference about predictors rather than correlations. The correlational analyses are correct as far as I can tell, but I think in some cases a regression analysis would be helpful to examine the unique contribution of, for example, the different aspects of identity in certain relations.

Finally, I would suggest to consider addressing some conceptual and theoretical concerns in the introduction and discussion. In the introduction, I would suggest to address in more depth what the expectations are for the identification processes of opponents of vaccines. Based on the current overview of the literature, it seems that it is already clear that anti-science is not central to their ingroup or beliefs. This would make answering the question, like it is done in the current study, superfluous. Another thing I think would be important for the discussion, in what “belief in science” or “trust in science” would mean for either group. This might be hard to answer based on the measures in the study, but I could imagine that each group has “their own science” to trust, meaning that the absolute level of trust in science is less relevant than which scientific findings are believed by either group.

Minor points

The methods section mentions that the questionnaire consists of three parts, but only two are explained

I think there is a slight mismatch in the number of participants between abstract and methods section.

The second hypothesis is not formally tested: it states that the correlation between level of identity and acceptance of science is stronger in the pro-vaccine group. The findings are presented in Table 4, but no conclusion is drawn about the difference between the pro- and anti-vaccine groups.

The writing is generally clear. Some expressions or phrases need to be made more consistent or clear, such as the “predictors”, “mechanisms” that I mentioned before, but also a phrase like “another question that arises” (p.5) needs some substantiation: where does this arise from exactly? Other than that, I thought that the language and writing was clear.

Reviewer #2: This well-written article explores group identity in pro- and anti- vaccination individuals who self-identify as being active in conversations about this issue. It finds that pro-vaccination participants have stronger group identity, both groups identify as pro-science more than not, and both take a dim view of the other’s scientific basis. With some adjustments, I think this article would be a good addition to the literature.

I note that the abstract does not really pull out the most interesting findings and urge the authors to edit it to play to the article’s strengths.

The authors build from a basis of thinking about vaccinators and non-vaccinators as people who engage in (non)scientific thinking, but at line 97-99 this appears somewhat jarring as a basis for the study, even though the authors have offered some basis for why we might think this. I think the reason it doesn’t feel right here is that pro-vax supporters are assumed to be pro-science and that this is a causal factor for their support for vaccination. I will return to this topic later. For now, I think these lines can be improved by offering evidence as to why we might believe that pro-vaxxers are also passionately pro-science. There may be academic articles that cover this point, and the authors should look for some. Failing that, they could rest on the evidence that many skeptic organisations (who are first and foremost science lovers!) engage quite heavily with the vaccination issue, including through podcasts, articles etc, and cite some examples. I feel that this evidence would make the authors’ assertion that science views matter to pro-vax views more convincing, and hence make the study’s premise more convincing.

I note that I’m a qualitative researcher so I cannot comment on the article’s quant methods. It did strike me that the N is small, and I’d like the authors to defend in the article whether the study is sufficiently powered.

Because I’m not a psychologist, I prefer explanations of results that non-quant people can understand. So, for example, at lines 212, I’d like the sentence about significant differences to actually explain who thought what. Likewise at line 221, please explain what this means in concrete language about the people in your study. At line 260, please add something that demonstrates what it means (you talk about attitudes toward science but not in which direction) Eg “meaning that pro-science people had stronger group identities, whether they were pro or anti vaccine.” This is the kind of concrete language I’m asking for throughout. At line 262, again please explain in concrete terms what the first half of this sentence is saying about your population. Including a basic explanation of what your findings actually mean in concrete terms will make your article more readable and probably more likely to be cited by others outside your discipline.

At line 290 , once again we revisit the idea that pro science motivates pro- vax behaviour / attitude / action. I’d like the authors to consider, and include in their limitations, the idea that it might swing the other way. It could also be that pro-vax people get socialised into being strong supporters of science! I raise this point because this was my journey; it might well be that of other people. Consider that causation runs the other way.

Line 313, you find that there was not a factor that clearly unites vaccine rejecters. This has been discussed in other literature – see

Kahan, D. M. 2014. Vaccine Risk Perceptions and Ad Hoc Risk Communication: An Empirical Assessment. In CCP Risk Perception Studies: Yale Law School; Harvard University - Edmond J. Safra Center for Ethics.

And then see critiques that he, too, might not have looked for the right thing:

Attwell, K., and Smith, D. T. 2017. Parenting as Politics: Social Identity Theory and Vaccine Hesitant Communities. International Journal of Health Governance 22 (3):183-198.

For your reflections on how the groups view each other’s scientific knowledge at lines 326-332, some other studies may be useful:

Attwell, K., Smith, D. T., and Ward, P. R. 2018. ‘The Unhealthy Other’: How vaccine rejecting parents construct the vaccinating mainstream. Vaccine 36 (12):1621-1626. doi: https://doi.org/10.1016/j.vaccine.2018.01.076.

Rozbroj, T., Lyons, A., and Lucke, J. 2019. The mad leading the blind: Perceptions of the vaccine-refusal movement among Australians who support vaccination. Vaccine 37 (40):5986-5993. doi: https://doi.org/10.1016/j.vaccine.2019.08.023.

And for your discussion of the motivations of anti-vaccinators lines 338-340, these may be useful:

Ward, P. R., Attwell, K., Meyer, S. B., Rokkas, P. R., and Leask, J. 2017. Understanding the perceived logic of care by vaccine-hesitant and vaccine-refusing parents: A qualitative study in Australia. PLoS One 12 (10).

Reich, J. 2014. Neoliberal Mothering and Vaccine Refusal: Imagined Gated Communities and the Privilege of Choice. Gender & Society 28 (5):679-704. doi: 10.1177/0891243214532711.

All suggestions are made for the authors’ benefits with no expectations that they cite any of them unless they consider them useful for their framing and wider analysis.

Reviewer #3: The paper provides a helpful insights to an important topic, which hopefully will have a positive impact in both practical and theoretical aspects. I was thrilled by the decision of the researcher to adopt a group dynamic approach to the topic, which does provide more depth of understanding. I have no major comments on the manuscript, except one concern in regard how the researchers talk about group identity, as following:

The researchers seem to base their approach on assumption that science is “the” fundamental force behind the position toward vaccines and activism around it. Thus, they measured attitude towards science and science-based social identity, and used them in their analysis. Although the authors believe that “any domain (value, symbol, attitude) can become the basis for group identity, provided that it is accepted and considered valid by the members of the group” (P.4, Line 91) they seemed to be over focused on science alone as base for group membership, and ignoring other possible sources, which might explain some possible flaws in the analysis and conclusions. Other possible sources for group identity can be the individual liberal rights (resisting the tyranny of state), which is known to be a common part of anti-vaccine debates. It is true that the paper makes an excellent argument that anti-vaccine groups are not necessarily have a common negative attitude toward science , and show in the result that such attitude seems not to be the base for their group identity; However, I’m concerned about framing such results as denying the role of social identity, in general. So, if the authors aims to explore specifically the role of science-based social identity, then I would suggest making that clear, wherever needed, to avoid misleading conclusions such as “social identity of vaccination rejecters is weaker than that of vaccination supporters”.

6. PLOS authors have the option to publish the peer review history of their article (what does this mean?). If published, this will include your full peer review and any attached files.

Reviewer #1: No

Reviewer #2: No

Reviewer #3: No

---

## [Author Response · Author response to Decision Letter 0]

24 Jun 2021

Dear Editor and Reviewers,

Thank you for the insightful reviews. Below are our responses to each comment.

Editor’s comments:

R1 and R2 are concerned that the rationale is under-developed and does nor motivate the hypotheses.

We extensively modified the introductory part of the manuscript, including a separate part with hypotheses and their justification.

R2 also points out that the abstract needs to better represent the study

We re-wrote the abstract to better indicate what the research is about. 

Language and style needs some work

We changed the style in many parts of the manuscript and hope that the language is better now; extensive changes requested by the Reviewers forced us to rewrite parts of the manuscript from scratch and we hope these parts are also more readable.

Please elaborate on Ariadna Nationwide Research Panel, e.g. where, which country, public daya or not?

We provided a detailed explanation in the Method section

Provide full list of items as supplement if you have not done so already.

We added a reference to Roccas’ social identity scale in the text. Since this is not our own scale, we don’t feel it would be ethical to reproduce it in full. All other questions are presented in full within the Methods section.

Detail the software you used

We added information about using SPSS for analyses into the Results section

Figures appeared to be directly copied and pasted from SPSS output?

We changed the figures to ones produced outside SPSS.

More importantly, consider running additional analyses to determine which aspect of group membership is more influential on attittudues toward science and vaccination

We ran regression analyses to determine which aspects of group identity are most important for attitudes toward science, with interesting results.

 Please elaborate on the limitations of your research. This is important.

We added a paragraph about study limitations to the Discussion.

Reviewer 1’s comments:

I found the overall argumentation and structure of the article rather difficult to follow. Although several aims and hypotheses were described, I did not find a clear research question in the manuscript. Additionally, there were some inconsistencies between hypotheses, measures, and the reported findings; most notably around the concept of social identity. I will elaborate on these concerns below.

We extensively rewrote the introduction and discussion and hope that the paper is now easier to follow.

Regarding the use of social identity in the current manuscript, some things did not become clear to me. My main concern is that the manuscript does not provide a clear reason why and how group identity/social identity should play a role among vaccine supporters and opponents. I am sure there are plenty of good reasons to study group processes here, but without a clear reason how it needs to be studied, I found it difficult to assess whether the methods were appropriate to do so. Could, for example, being personally pro- or anti vaccines not explain the same patterns in support for science ?

We added more justification for investigating the relation between group identity and vaccine-related attitudes. The question whether being ‘personally pro/anti vaccine explain the same patterns’ is quite interesting and we cannot answer it based on our study; however we did use the well-established social identity theory, along with the social identity scale, to investigate social identity of pro-vaccine and anti-vaccine individuals, so there is very little discrepancy between theory and method. The main goal was to use this established theory to investigate the particular group of pro-vaccine and anti-vaccine people.

The methods to assess social identification match the useful distinction between the four aspects of group identity (p.4). However, two things are unclear or inconsistent here. First, the hypotheses mention “level of group identity” as a singular measure, whereas the correlations are presented for each of the four aspects of group identity. Moreover, the single reliability coefficient in the measures section suggests that indeed all 16 items are considered as a single group identity scale. When considered separately, I think a factor analysis would add to the understanding of these four sub-scales.

We changed the wording to ‘levels’. Also, we added alpha coefficients for all subscales. 

 Second, the methods section lacks sample items for the identity scale. Based on the data set, I could not fully assess what the group identity exactly pertains clearly to in the items. I think this is especially important given my earlier point that it is unclear what the group identity adds in explanatory power over and above personally being pro- or anti vaccines.

We refrained from providing a full set of items for Roccas’ scale, since it is an existing scale by another author and we don’t have the right to fully reproduce it. We did add a few sample items into the Methods section.

In the theory, I also would have liked to a see a specific discussion on this: what does a pro- or anti vaccine identity look like exactly?

We have developed, as far as possible on the basis of our research, the characteristics of the group identity of both groups

I think the sub-sample drawn from the larger representative sample is fitting to test the hypotheses. The selection criteria are well-argued in the methods section. Perhaps the introduction could have been clearer that the current study concerns active/involved opponents and supporters of vaccines.

We added a more clear statement that this is the case

Based on vaccination rates, vaccine supporters can either be an overwhelming majority group or form a minority and this can influence identification processes. Some more background information on this would help argue for the current hypotheses: how do the opponents and supporters compare to the general representative sample that they were drawn from? And what can be said beforehand about pro- and anti-vaccine groups in the current context, both practically and theoretically?

We included these statistics in the description of our sample. Indeed, pro-vaccine individuals were thankfully a vast majority (nearly 75%), and anti-vaccine individuals were only 5% of the sample. The final sample size was constricted by the numer of anti-vaccine participants who were also active, which was about a fourth of the total anti-vaccine sample. While this may suggest that anti-vaccine individuals feel a stronger association with their minority group, these two groups operate mostly online in information bubbles, so anti-vaccine individuals may as well feel as if their group is larger than it actually is. This is why we did not state any theoretical assumptions regarding their identity based on just one group being larger or smaller than the other. 

The methods and results sections contained some other puzzling information. I found the types of involvement very useful in order to get a better understanding of who the participants in the study were. However, the type of involvement was not mentioned in the introduction, hypotheses, nor methods and therefore came as a surprise to me. In the results and discussion, involvement seems to be discussed as a predictor. Additionally, the “outgroup identity” in Table 2 was not clear to me. I did not find this measure in the methods section.

Thank you for this comment. Commitment was an important criterion for selecting proponents and opponents of vaccination for the study. We have now devoted more to this factor in the introduction, method and hypotheses. As for ‘outgroup identity’, we included this measure by mistake, it has now been removed from the manuscript (It was an attempt at measuring how the two groups perceive each other’s identities, but it does not fit into the theoretical framework of this paper).

For the general methodological approach of the manuscript, I think the language in the abstract, introduction, and hypotheses is not always consistent. For example, the abstract mentions “mechanisms”, suggesting group identity to potentially play a mediating role that is not test. On page 2, “understanding the predictors of […] attitudes towards vaccination” is mentioned, suggesting at least some kind of inference about predictors rather than correlations.

We thoroughly rewrote the manuscript to avoid this type of language, as we meant ‘mechanisms’ in a theoretical context and not as a mediator; we also conducted regression analyses which allow us to speak of ‘predictors’ now, but this assumption still feels too strong.

The correlational analyses are correct as far as I can tell, but I think in some cases a regression analysis would be helpful to examine the unique contribution of, for example, the different aspects of identity in certain relations.

We conducted regression analyses and added them to the manuscript, as their results are indeed very interesting.

Finally, I would suggest to consider addressing some conceptual and theoretical concerns in the introduction and discussion. In the introduction, I would suggest to address in more depth what the expectations are for the identification processes of opponents of vaccines. Based on the current overview of the literature, it seems that it is already clear that anti-science is not central to their ingroup or beliefs. This would make answering the question, like it is done in the current study, superfluous. Another thing I think would be important for the discussion, in what “belief in science” or “trust in science” would mean for either group. This might be hard to answer based on the measures in the study, but I could imagine that each group has “their own science” to trust, meaning that the absolute level of trust in science is less relevant than which scientific findings are believed by either group.

We added more commentary on this in the Discussion, however it does not seem possible to clearly answer these issues with our data.

The methods section mentions that the questionnaire consists of three parts, but only two are explained

We added a clear indication that ‘attitudes toward science’ and ‘perceived scientific knowledge’ were the third part.

I think there is a slight mismatch in the number of participants between abstract and methods section.

Thank you for noticing this, we amended the mistake

The second hypothesis is not formally tested: it states that the correlation between level of identity and acceptance of science is stronger in the pro-vaccine group. The findings are presented in Table 4, but no conclusion is drawn about the difference between the pro- and anti-vaccine groups.

We modified the hypothesis to reflect what we initially meant more precisely.

 a phrase like “another question that arises” (p.5) needs some substantiation: where does this arise from exactly?

We changed this wording so it does not create confusion.

Reviewer 2’s comments:

 I note that the abstract does not really pull out the most interesting findings and urge the authors to edit it to play to the article’s strengths.

We thoroughly rewrote the abstract to make it reflect the study better

I think the reason it doesn’t feel right here is that pro-vax supporters are assumed to be pro-science and that this is a causal factor for their support for vaccination.

Thank you for noticing this; the relation between pro-science and vaccination support can go either way, we changed the wording to reflect it.

It did strike me that the N is small, and I’d like the authors to defend in the article whether the study is sufficiently powered.

The small N is a result of the anti-vaccine group being, thankfully, only a small percentage of the population (in our research it was only 5% of the primary sample, we added these statistics to the text) and an even smaller percentage are active anti-vaccine individuals (about 25% of the entire anti-vaccine group). The base sample for the study was over 10500 participants, and the low final Ns show how few active anti-vaccine people there actually are. 

Because I’m not a psychologist, I prefer explanations of results that non-quant people can understand.

While we are used to presenting results in standard APA style, we tried our best to explain results in concrete terms wherever possible. However, we felt that it is necessary to keep the standard wording alongside, not to draw critique from ‘quant people’.

we revisit the idea that pro science motivates pro- vax behaviour / attitude / action. I’d like the authors to consider, and include in their limitations, the idea that it might swing the other way. It could also be that pro-vax people get socialised into being strong supporters of science!

Of course this argument is very valid; the tendency to assume that pro-science causes vaccine support seems to stem from existing research on ‘anti-scientific beliefs’, however we agree that especially for vaccine rejection and support the relation may go the other way – people are known to reject vaccines and at the same time support science, even though this science becomes very distorted with time. We changed our wording to less causal.

Line 313, you find that there was not a factor that clearly unites vaccine rejecters. This has been discussed in other literature

Thank you for the valuable literature suggestions. We used some of the cited papers, mainly to support the idea that vaccine rejection may have various causes.

Reviewer 3’s comments:

The researchers seem to base their approach on assumption that science is “the” fundamental force behind the position toward vaccines and activism around it. Thus, they measured attitude towards science and science-based social identity, and used them in their analysis. Although the authors believe that “any domain (value, symbol, attitude) can become the basis for group identity, provided that it is accepted and considered valid by the members of the group” (P.4, Line 91) they seemed to be over focused on science alone as base for group membership, and ignoring other possible sources, which might explain some possible flaws in the analysis and conclusions.

It seems that we may have written some sections of the article in a way that caused a misunderstanding. We did not assume that pro-science or anti-science is the only cause for vaccine support or rejection. Especially we did not assume that attitudes toward science are the basis of group identity. Our main assumption was in fact that being pro-vaccine or anti-vaccine causes one to perceive themselves as part of a larger group, and identifying themselves with it. Group identity of pro-vaccine and anti-vaccine group members was our main focus of research. Whether these groups exhibit pro-scientific or anti-scientific beliefs was more of a side question which could explain people being pro-vaccine/anti-vaccine, as well as identifying themselves with each group in different ways. We changed the order of most of the paper to better indicate it, and also indicated some other possible sources of anti-vaccine beliefs. 

It is true that the paper makes an excellent argument that anti-vaccine groups are not necessarily have a common negative attitude toward science , and show in the result that such attitude seems not to be the base for their group identity; However, I’m concerned about framing such results as denying the role of social identity, in general. So, if the authors aims to explore specifically the role of science-based social identity, then I would suggest making that clear, wherever needed, to avoid misleading conclusions such as “social identity of vaccination rejecters is weaker than that of vaccination supporters

We aimed to explore the group identity (or group identification) of pro-vaccine and anti-vaccine people. It could be said that we wanted to specifically explore vaccine-based identity, attitudes toward science being one of its particular aspects (to test the hypothesis that vaccine support and vaccine rejection are caused by science support and science rejection, respectively, which turned out to be untrue. Thank you for pointing out that our paper can be interpreted in this way; we changed the wording of statements which we feel may have suggested that science is the only or main basis for group identity of vaccine supporters and rejecters.

---

## [Decision Letter · Decision Letter 1]

11 Oct 2021

PONE-D-21-02562R1

Active pro-vaccine and anti-vaccine groups: their group identities and attitudes toward science.

PLOS ONE

Dear Dr. Polak,

Thank you for submitting your manuscript to PLOS ONE. After careful consideration, we feel that it has merit but does not fully meet PLOS ONE’s publication criteria as it currently stands. Therefore, we invite you to submit a revised version of the manuscript that addresses the points raised during the review process. While one reviewer recommend accept, the other recommended substantial changes to nearly all aspects of the paper from tiny details to big picture issues.

We look forward to receiving your revised manuscript.

Kind regards,

Peter Karl Jonason

Academic Editor

PLOS ONE

Reviewers' comments:

Reviewer's Responses to Questions

**Comments to the Author**

1. If the authors have adequately addressed your comments raised in a previous round of review and you feel that this manuscript is now acceptable for publication, you may indicate that here to bypass the “Comments to the Author” section, enter your conflict of interest statement in the “Confidential to Editor” section, and submit your "Accept" recommendation.

Reviewer #1: (No Response)

Reviewer #2: All comments have been addressed

2. Is the manuscript technically sound, and do the data support the conclusions?

Reviewer #1: Partly

Reviewer #2: (No Response)

3. Has the statistical analysis been performed appropriately and rigorously? 

Reviewer #1: I Don't Know

Reviewer #2: (No Response)

4. Have the authors made all data underlying the findings in their manuscript fully available?

Reviewer #1: Yes

Reviewer #2: (No Response)

5. Is the manuscript presented in an intelligible fashion and written in standard English?

Reviewer #1: Yes

Reviewer #2: (No Response)

6. Review Comments to the Author

Reviewer #1: In the revised manuscript “Active pro-vaccine and anti-vaccine groups: their group identities and attitudes toward science”, most of the suggestions made by reviewers seem to be addressed in one way of another. Additionally, the relevance of the topic of attitudes towards vaccination remains an important one. Nonetheless, I do think that the structure of the manuscript still contains omissions between the theoretical rationale, hypotheses, and reported findings.

One of the problems that is still in the manuscript, is how “level of involvement” is treated. First, the findings on level of involvement are reported quite extensively in section 3.2 of the results. Also, on line 198, it says that there is an aim of investigating this variable in relation to group identity. However, there is no specific theory, hypothesis, nor a clear measure for it. Moreover, because the participants are selected on this variable, I think section 3.2 can logically only be used as some kind of description of either group. The conclusion on page 19 (in the discussion) seems to indicate that this was the intended way of using level involvement as well.

Another problematic part of the manuscript remains the role of social identity and how it is operationalized. The reasons why social identity is used is not entirely clear even in the revised manuscript, especially when it comes to the four dimensions of social identity. In the expectations in the theoretical framework, but also in the hypothesis itself, there is no mention of difference between any of the four types of identification. Yet, these are measured and reported separately in the methods and results section. I find the regression analysis of the various social identity measures interesting but also would like to know how strongly they are related. Additionally, ideally there would be some kind of control variable for religiosity in these analyses in order to rule out alternative explanations for attitudes towards science. I do not know whether such additional measures are available.

Additionally, I think there is still work to be done in order to completely make the case for what the identities are that are measured; what their content could be. For example, some of the individual differences that are mentioned on page 3 include orthodox religiousness and moral purity concerns, both of which could easily be part of shared ingroup norms and not individual differences. The authors do not make clear how, then, the intergroup relations perspective adds to this line of research. One solution for this, I suggest, would be to provide context as to who individuals in either group could be, given the current sociopolitical situation in Poland. On page 6, there is some information on why the anti-vaccine group might have more diverse motivations and therefore identify less as anti-vaccine supporters. In many contexts, this might seem counter-intuitive because the majority group in most situations would identify less strongly. I urge the authors to expand on this point of view by using the national context and existing literature. The Rutjens et al article on line 168 might be useful here, but then some reflection on how the current study adds to these findings is important.

I would recommend changing the structure of the theoretical framework to some extent, in order to be able to see the main hypotheses and their argumentation sooner. For example, I find the hypothesis about perception of anti-science attitudes of the other group interesting, but there is barely any theorizing on it. The few lines around line 212 mention “basic rules of social cognition” while only when discussing the hypotheses, the theoretical foundations of outgroup bias are mentioned (line 231) but not discussed in depth.

All in all, quite a few confusing aspects of the initial manuscript are resolved in this revision and the correlates of pro- and anti-vaccine individuals are reported clearer now. Nonetheless, my points above still make it difficult sometimes to assess the necessity and purpose of some of the measures.

Minor issues

Space missing on line 69 and line 128

The reference on opinion-based groups should maybe already been mentioned on line 136.

The discussion paragraph starting at line 459 invokes all kinds of motivations for group membership, but I suggest making clearer what is known from previous literature versus what the findings of the current research are.

The discussion does not engage much with the theory: the findings are explained well, but it is not clear what it means for existing theory.

Reviewer #2: xxxxxxxxxxxxxxxxxxxxxxxxxxxxxxxxxxxxxxxxxxxxxxxxxxxxxxxxxxxxxxxxxxxxxxxxxxxxxxxxxxxxxxxxxxxxxxxxxxxxxxxxxxxxxxxxxxxxxxxxxxxxxxxxxxxxxxxxxxxxxxxxxxxxxxxxxxxxxxxxxxxxxxxxxxxxxxx

7. PLOS authors have the option to publish the peer review history of their article (what does this mean?). If published, this will include your full peer review and any attached files.

Reviewer #1: No

Reviewer #2: No

---

## [Author Response · Author response to Decision Letter 1]

22 Nov 2021

Reviewer #1: In the revised manuscript “Active pro-vaccine and anti-vaccine groups: their group identities and attitudes toward science”, most of the suggestions made by reviewers seem to be addressed in one way of another. Additionally, the relevance of the topic of attitudes towards vaccination remains an important one. Nonetheless, I do think that the structure of the manuscript still contains omissions between the theoretical rationale, hypotheses, and reported findings.

One of the problems that is still in the manuscript, is how “level of involvement” is treated. First, the findings on level of involvement are reported quite extensively in section 3.2 of the results. Also, on line 198, it says that there is an aim of investigating this variable in relation to group identity. However, there is no specific theory, hypothesis, nor a clear measure for it. Moreover, because the participants are selected on this variable, I think section 3.2 can logically only be used as some kind of description of either group. The conclusion on page 19 (in the discussion) seems to indicate that this was the intended way of using level involvement as well.

 Thank you for pointing out that our description of of „level of involvement” may be unclear. The basis for referring to this construct is the contemporary approach to group formation, in which involvement is treated as a fundamental process (Moreland and Levine, 1994). In addition to the question of defining oneself in terms of belonging to a group, contemporary analyses place a strongly emphasis on different forms of involvement in group functioning (Hogg, 1992). Therefore, in our study, the level of involvement in various activities was the criterion for selecting people into the study groups (in addition to strong pro- and anti-vaccine beliefs). Whereas in the paragraph concerning the aims of the study (on lines 189-190) we stated that the level of involvement was one of the study variables, declaring that the aim of our study was, among other things, to test its relationship with group identity and attitudes towards science. In fact, it was not a variable, but a selection criterion so the word ‘relationship’ was misleading. In the current manuscript, we have removed this phrase and also referenced the above view on the importance of involvement in group functioning.

Another problematic part of the manuscript remains the role of social identity and how it is operationalized. The reasons why social identity is used is not entirely clear even in the revised manuscript, especially when it comes to the four dimensions of social identity. In the expectations in the theoretical framework, but also in the hypothesis itself, there is no mention of difference between any of the four types of identification. Yet, these are measured and reported separately in the methods and results section. I find the regression analysis of the various social identity measures interesting but also would like to know how strongly they are related. 

We have tried to better clarify the reasons for using social identity according to Roccas, and we have added more information in the manuscript about the four dimensions of social identity and the possible differences and relationships between them. Based on the results of our study, we performed additional analyses on the relationship between these dimensions of social identity (correlations between identity modes and expanded CFA).

Additionally, ideally there would be some kind of control variable for religiosity in these analyses in order to rule out alternative explanations for attitudes towards science. I do not know whether such additional measures are available.

Unfortunately, religiosity of the participants in our study was not measured. While we understand that measuring religiosity would add a possible additional level of explanation, so would measuring other variables known to relate to vaccine rejection and/or anti-science, which however would require a much larger research project, whereas we focused on group identity from the beginning.

Additionally, I think there is still work to be done in order to completely make the case for what the identities are that are measured; what their content could be. For example, some of the individual differences that are mentioned on page 3 include orthodox religiousness and moral purity concerns, both of which could easily be part of shared ingroup norms and not individual differences. The authors do not make clear how, then, the intergroup relations perspective adds to this line of research. One solution for this, I suggest, would be to provide context as to who individuals in either group could be, given the current sociopolitical situation in Poland. On page 6, there is some information on why the anti-vaccine group might have more diverse motivations and therefore identify less as anti-vaccine supporters. 

As stated above, we have added more information about the four modes of group identity. Thank you for pointing out that the individual variables mentioned can be linked to norms, which are an important part of the structure of a group.

We understand that the study of group processes covers an extremely wide range of phenomena. One can focus, among other, on the individual cognitive and motivational processes leading to group behavior, on the interconnection of individual, interpersonal and social processes, on the cohesiveness of the group and its identity, etc. (Moreland & Levine, 1994).

In the manuscript, we mentioned traditional research on seeing which individual variables in a population might be predictors of attitudes toward science, toward vaccination, or toward climate change (e.g., whether climate change denial is associated with political conservatism). In contrast, we purposively recruited two groups for the study that had to meet not only the criterion of acceptance or rejection of vaccinations but also the criterion of engagement in activity related to these attitudes - to describe and compare an important group process such as group identity. In short – while existing research investigates individual differences which may be associated with group norms or group behavior (which is an assumption worth researching by itself), we investigated how much and in what manner the active pro- and anti-vaccine individuals identify with the groups they belong to – a key factor to understand whether their attitudes and beliefs are individual in nature or whether they truly identify with their respective groups

The proposition to include the current sociopolitical situation in Poland is interesting, as polls indicate that being anti-vaccine is common in supporters of the ruling Law and Justice party (a conservative, populist party), and that there is a general distrust in the government typical for the Eastern Bloc, which fuels vaccine hesitancy. However, we refrained from discussing this in our manuscript, as we did not directly measure political attitudes, so it would be pure speculation. In truth, we are not aware of any research directly describing the ‘active’ vaccine supporters and rejecters in Poland, and these groups may be different from the general anti-vaccine and pro-vaccine demographics.

In many contexts, this might seem counter-intuitive because the majority group in most situations would identify less strongly. I urge the authors to expand on this point of view by using the national context and existing literature. The Rutjens et al article on line 168 might be useful here, but then some reflection on how the current study adds to these findings is important.

We confirm the above remark. The minority group usually has a stronger group identity than a majority group; especially with regard to uncontrollable characteristics (e.g. skin color, nationality). But this does not apply to the anti-vaccine group, which is probably because people rarely think of themselves in terms of pro-vaccinationist vs. anti-vaccinationist. If it happens to them (e.g. in the course of internet discussions), pro-vaccinationists see the homogeneity of their group (everyone believes in science) which strengthens the sense of group identity, while anti-vaccinationists see the heterogeneity of their group (some of them believe in science, others do not; some are afraid of epidemics, others do not; some believe in conspiracy, others do not). We added this reasoning to the discussion of our research findings.

I would recommend changing the structure of the theoretical framework to some extent, in order to be able to see the main hypotheses and their argumentation sooner. For example, I find the hypothesis about perception of anti-science attitudes of the other group interesting, but there is barely any theorizing on it. The few lines around line 212 mention “basic rules of social cognition” while only when discussing the hypotheses, the theoretical foundations of outgroup bias are mentioned (line 231) but not discussed in depth.

We changed the structure of the paper and added more theoretical background.

All in all, quite a few confusing aspects of the initial manuscript are resolved in this revision and the correlates of pro- and anti-vaccine individuals are reported clearer now. Nonetheless, my points above still make it difficult sometimes to assess the necessity and purpose of some of the measures.

Minor issues

Space missing on line 69 and line 128

The reference on opinion-based groups should maybe already been mentioned on line 136.

The discussion paragraph starting at line 459 invokes all kinds of motivations for group membership, but I suggest making clearer what is known from previous literature versus what the findings of the current research are.

The discussion does not engage much with the theory: the findings are explained well, but it is not clear what it means for existing theory.

We spell-checked the entire manuscript, and hope to have solved all problems including missing spaces. As regards the discussion, unfortunately there is very little theory and the research is very exploratory in nature; to our knowledge, there has been no research on the identity of pro- and anti-vaccine groups, let alone strictly active ones. We added clearer statements that most of the discussion refers directly to our research. We also added some suggestions on how to merge our research with existing theory, and especially how the results of our research differ from what would be expected based on earlier research on group identity, for instance that the minority may not have a stronger identity than the majority.

---

## [Editor Report · Decision Letter 2]

9 Dec 2021

Active pro-vaccine and anti-vaccine groups: their group identities and attitudes toward science.

PONE-D-21-02562R2

Dear Dr. Polak,

We’re pleased to inform you that your manuscript has been judged scientifically suitable for publication and will be formally accepted for publication once it meets all outstanding technical requirements.

Kind regards,

Peter Karl Jonason

Academic Editor

PLOS ONE
---

## [Editor Report · Acceptance letter]

13 Dec 2021

PONE-D-21-02562R2 

*Active pro-vaccine and anti-vaccine groups: their group identities and attitudes toward science.*

Dear Dr. Polak:

I'm pleased to inform you that your manuscript has been deemed suitable for publication in PLOS ONE. Congratulations! Your manuscript is now with our production department. 

Kind regards, 

on behalf of

Dr. Peter Karl Jonason 

Academic Editor

PLOS ONE